# Transnuclear mice reveal Peyer's patch iNKT cells that regulate B-cell class switching to IgG1

Eleanor Clancy-Thompson[1,‡], Gui Zhen Chen[1,‡], Nelson M LaMarche[2,3], Lestat R Ali[1] (ID), Hee-Jin Jeong[1,†], Stephanie J Crowley[1], Kelly Boelaars[1,4], Michael B Brenner[2,3], Lydia Lynch[2,3] & Stephanie K Dougan[1,3,*] (ID)

## Abstract

Tissue-resident iNKT cells maintain tissue homeostasis and peripheral surveillance against pathogens; however, studying these cells is challenging due to their low abundance and poor recovery from tissues. We here show that iNKT transnuclear mice, generated by somatic cell nuclear transfer, have increased tissue resident iNKT cells. We examined expression of PLZF, T-bet, and RORγt, as well as cytokine/chemokine profiles, and found that both monoclonal and polyclonal iNKT cells differentiated into functional subsets that faithfully replicated those seen in wild-type mice. We detected iNKT cells from tissues in which they are rare, including adipose, lung, skin-draining lymph nodes, and a previously undescribed population in Peyer's patches (PP). PP-NKT cells produce the majority of the IL-4 in Peyer's patches and provide indirect help for B-cell class switching to IgG1 in both transnuclear and wild-type mice. Oral vaccination with α-galactosylceramide shows enhanced fecal IgG1 titers in iNKT cell-sufficient mice. Transcriptional profiling reveals a unique signature of PP-NKT cells, characterized by tissue residency. We thus define PP-NKT as potentially important for surveillance for mucosal pathogens.

**Keywords** IL-4, tissue-resident iNKT cells; oral vaccines; Peyer's patches; transnuclear mice

**Subject Categories** Immunology

**The EMBO Journal (2019) 38: e101260**

## Introduction

iNKT cells are T cells with semi-invariant TCRs that recognize lipid antigens presented on CD1d. They exist as a pre-expanded pool and can rapidly respond by producing a range of different cytokines (Brennan *et al*, 2013). iNKT cell functional subsets have been described that parallel the CD4 T-cell subsets: NKT1 cells express T-bet and are poised to secrete IFNγ; NKT2 cells express high levels of PLZF and are poised to secrete IL-4, while NKT17 cells are RORγt[+] and poised to secrete IL-17 (Kim *et al*, 2015; Wang & Hogquist, 2018). Each of these subsets can be found in the thymus and appear at different ratios in spleen and liver, which are the most abundant sources of iNKT cells in the mouse (Engel *et al*, 2016; Tuttle & Gapin, 2018).

iNKT cells are also found in disparate tissues such as lung, adipose tissue, and intestinal lamina propria (Crosby & Kronenberg, 2018). In lung, iNKT cell production of GM-CSF helps control *Mycobacterium tuberculosis* infection (Rothchild *et al*, 2017). iNKT cells in the gut interact with CD1d on epithelial cells to cause feedback production of IL-10 under homeostatic conditions (Olszak *et al*, 2014), but can be activated by oxazolone-induced inflammation to trigger colitis (Heller *et al*, 2002; Iyer *et al*, 2018). Gut iNKT cells are also induced by microbial ligands early in life (Olszak *et al*, 2012; An *et al*, 2014) and help shape the nascent microbiome (Selvanantham *et al*, 2016; Saez de Guinoa *et al*, 2018). In adipose tissue, iNKT cell interactions with macrophages set the metabolic tone of the whole animal and affect insulin sensitivity and propensity toward obesity (Lynch *et al*, 2012, 2015; Exley *et al*, 2014). In addition to the well-described NKT1/2/17 subsets, iNKT cells can also have follicular helper function (Chang *et al*, 2011; King *et al*, 2011; Dellabona *et al*, 2014; Doherty *et al*, 2018) and regulatory function (Monteiro *et al*, 2010; Sag *et al*, 2014), or produce primarily IL-9 (Kim & Chung, 2013; Monteiro *et al*, 2015). The role of iNKT cell subsets in distinct tissue environments has been often difficult to elucidate due to poor cell recovery and a paucity of iNKT cells at baseline.

iNKT cells can provide B-cell help in two fashions: either by cognate interactions between CD1d-expressing B cells and CD40L-expressing iNKT cells (Galli *et al*, 2007; Barral *et al*, 2008; Leadbetter *et al*, 2008) or by non-cognate interactions whereby iNKT cells license dendritic cells to prime CD4 Tfh cells (Tonti *et al*, 2009; Vomhof-DeKrey *et al*, 2014). Cognate interactions generate short-term bursts of Ig production, but do not sustain long-term B-cell

1   Department of Cancer Immunology and Virology, Dana-Farber Cancer Institute, Boston, MA, USA
2   Department of Rheumatology, Brigham and Women's Hospital, Boston, MA, USA
3   Program in Immunology, Harvard Medical School, Boston, MA, USA
4   VU University Amsterdam, Amsterdam, The Netherlands
    *Corresponding author. Tel: +1 617 582 9609; Fax: +1 617 582 9610; E-mail: Stephanie_dougan@dfci.harvard.edu
    ‡These authors contributed equally to this work
    †Present address: Hongik University, Seoul, Korea

memory or generate long-lived plasma cells (King *et al*, 2011; Tonti *et al*, 2012; Vomhof-DeKrey *et al*, 2015). Non-cognate interactions do generate long-term memory, and several studies have shown differences between help provided by splenic iNKT Tfh versus CD4 Tfh, despite the fact that both cell types produce IL-21 and express CD40L (King *et al*, 2011; Tonti *et al*, 2012). In addition to cognate and non-cognate help, early production of IL-4 by iNKT cells in the lung was demonstrated to be critical for B-cell survival and entry into germinal centers upon infection with viral pathogens (Gaya *et al*, 2018). These studies defined iNKT cell provision of IL-4 as a third mechanism by which iNKT cells offer B-cell help (Gaya *et al*, 2018).

NKT cell functional differentiation can begin as early as thymic development (Lee *et al*, 2013). Signal strength through the TCR during positive selection can skew function, with higher affinity or more TCR signaling leading to NKT2 cells and lower TCR signaling required for NKT1 cell development (Matulis *et al*, 2010; Cruz Tleugabulova *et al*, 2016; Tuttle *et al*, 2018; Zhao *et al*, 2018). We previously reported a panel of iNKT cell transnuclear mice, cloned by somatic cell nuclear transfer from the nuclei of individual iNKT cells, which express monoclonal Vβ7 or Vβ8.2 TCRs (Clancy-Thompson *et al*, 2017). Tissue-specific factors have been implicated in iNKT cell subset specification, and our study unequivocally showed that monoclonal iNKT cells with different ligand specificities differentiate *in vivo* into all iNKT subsets at relatively normal frequencies, with a slight skewing of particular TCRs toward or away from NKT17 profiles. TCR specificity does not measurably affect localization of iNKT cells, their accumulation in tissues, or the expression of CD4 and has only a modest impact on transcription factor expression and cytokine production (Clancy-Thompson *et al*, 2017). Instead, tissue of origin plays a more dominant role in determining iNKT cell function, with iNKT cells from liver, skin-draining lymph nodes, spleen, and thymus having distinct cytokine and transcription factor profiles (Clancy-Thompson *et al*, 2017).

Given the importance of tissue-resident iNKT cells, we further investigated whether our panel of transnuclear mice could be used as an abundant source of tissue-resident iNKT cells. We here show that iNKT cells from mesenteric lymph node, skin-draining lymph node, adipose tissue, lung, liver, and spleen coordinate distinct cytokine profiles. These cytokine profiles are similar among polyclonal and each of our monoclonal lines, suggesting that TCR specificity plays a minor role in the differentiation of tissue-resident iNKT cells. Our transnuclear iNKT cells faithfully recapitulate the skewing of NKT1/2/17 ratios seen in disparate tissues from C57BL/6 mice, and transnuclear iNKT cells from adipose tissue are similar to those reported from C57BL/6 mice as well. Furthermore, we uncovered a novel population of iNKT cells residing in Peyer's patches and show that PP-iNKT cells are critical for B-cell class switching to IgG1$^+$ B cells in both steady state and upon oral vaccination.

## Results

### Tissue-resident iNKT cells are greatly enriched in iNKT transnuclear mice

We used somatic cell nuclear transfer to generate three independent lines of transnuclear (TN) mice, all of which use the identical

Vα14Jα18 TCRα chain, but with three distinct TCRβ rearrangements (Clancy-Thompson *et al*, 2017, 2018). When crossed to C57BL/6 mice, the TN TCR alleles segregate independently, which allowed us to establish a line of Vα14 TN mice that inherited only the rearranged TCRα locus and therefore develop polyclonal iNKT cells. These mice contain many-fold more iNKT cells in peripheral tissues than wild-type B6 mice (Fig 1A and B). The fold increase is especially pronounced in tissues where iNKT cells are rare, such as skin-draining and mesenteric lymph nodes, spleen, and lung.

To better profile cytokine and chemokine production by iNKT cells from different tissues, lymphocytes were harvested from spleen, mLN, sdLN, liver, adipose tissue, and lung of Vα14 and Jα18$^{-/-}$ mice (lacking iNKT cells), and cocultured with RAWd cells with or without α-GalCer for 24 h. Culture supernatants were then analyzed by cytokine bead array. Unfractionated lymphocyte populations were used; thus, the cytokines analyzed were not necessarily secreted by iNKT cells directly. To determine which cytokines and chemokines were produced in an iNKT cell-dependent manner, lymphocytes from Jα18$^{-/-}$ mice stimulated with α-GalCer were included as a negative control. As a second negative control, Vα14 lymphocytes were cultured in the absence of added antigen to determine the production of iNKT-dependent cytokines in response to endogenous ligands. Of the 31 analytes examined, 15 cytokines and chemokines showed iNKT cell-dependent production as defined by increased production in Vα14 cultures compared to Jα18$^{-/-}$ cultures across most tissues (Fig 1C). Mesenteric lymph node iNKT cells produced IL-4 and IL-13, as well as LIF and IL-2, suggesting that a large fraction of these cells are NKT2 (Fig 1C and Lee *et al*, 2015). Liver iNKT cells adopted more of an NKT1-like profile and produced CXCL9, CXCL10, and IFNγ. Liver iNKT cells also produced some IL-4, consistent with a previous report of IL-4 secretion by iNKT cells during sterile liver injury (Liew *et al*, 2017). Both inguinal LN and lung iNKT cells produced IL-17, although lung iNKT also produced IL-10 (Fig 1C). Adipose cultures produced IL-10, as previously reported (Lynch *et al*, 2012, 2015; Sag *et al*, 2014), as well as GM-CSF and eotaxin (Fig 1C). Spleen appeared to contain the most diverse iNKT population, capable of making nearly all cytokines and chemokines examined, although this likely reflects a mixture of several different functional subsets. Therefore, iNKT cells coordinate a signature cytokine profile dependent on the tissue of origin. The impact of tissue of origin on cytokine profiles was apparent in cultures containing polyclonal iNKT as well as monoclonal iNKT cells from Vβ7A, Vβ7C, and Vβ8.2 TN mice, although subtle differences in the relative magnitude of cytokine production from iNKT cells bearing different TCRs may exist (Fig EV1).

### Tissue-specific imprinting of TN iNKT cells recapitulates that seen in wild-type iNKT cells

To ensure that tissue-resident iNKT cells obtained from our TN mice faithfully recapitulate the phenotype of tissue-resident iNKT cells from wild-type mice, we examined expression of lineage-specific transcription factors in iNKT cells across multiple tissues in Vα14 and C57BL/6 mice (Fig 2A–F). IFNγ-poised NKT1 cells are characterized by expression of T-bet, while NKT2 cells are PLZF$^{high}$, and NKT17 cells express RORγt (Kim *et al*, 2015). Thymic differentiation is altered in iNKT TN mice, with an increase in the NKT2:NKT1 ratio in the thymus, consistent with our previous report (Clancy-

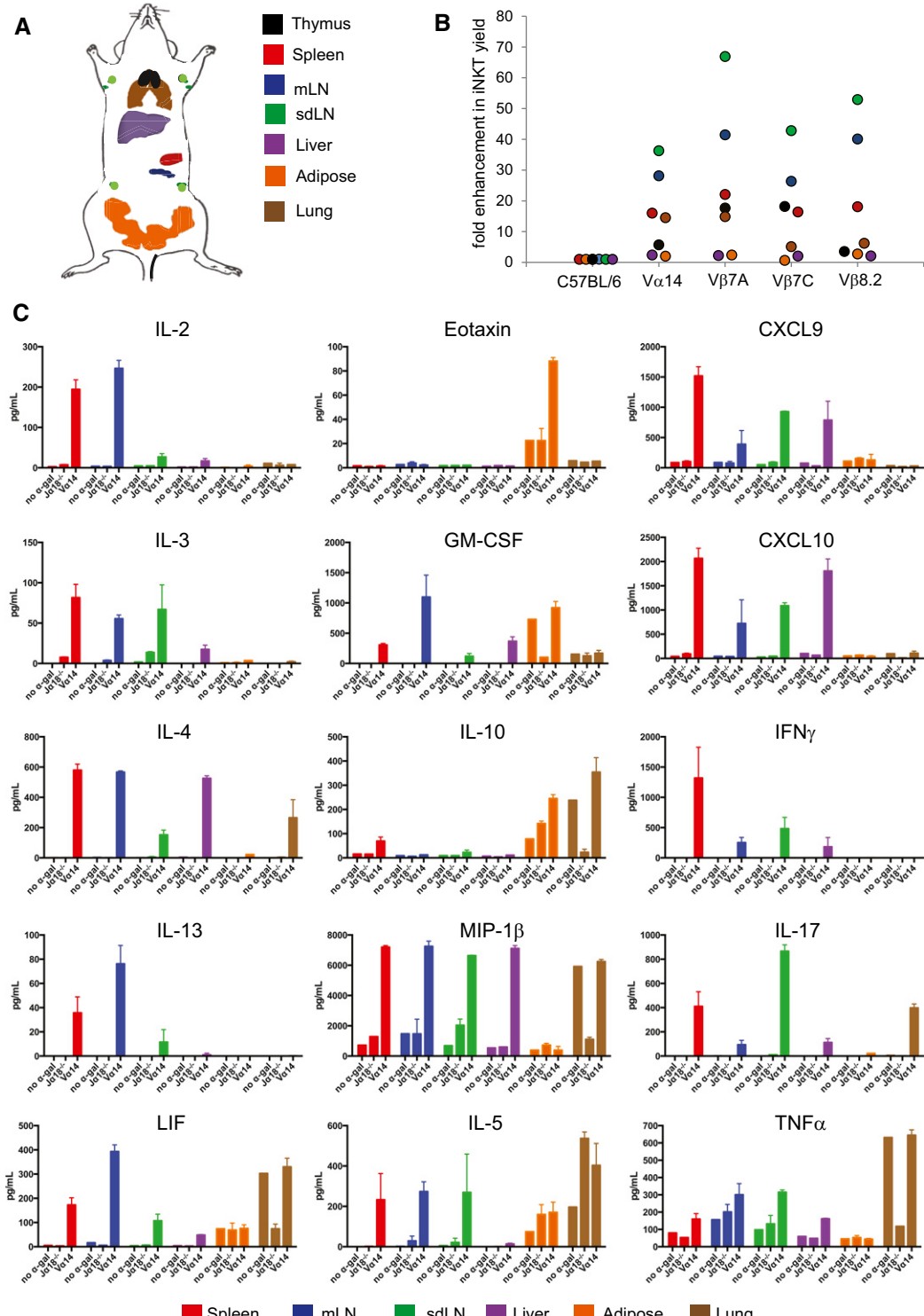

**Figure 1. iNKT TN mice have increased numbers of tissue-resident iNKT cells.**

A   Diagram representing the placement of various tissues analyzed for iNKT cells. mLN = mesenteric lymph node; sdLN = skin-draining lymph node.

B   Relative iNKT cell yield in various tissues from TN mouse lines compared to C57BL/6 mouse lines. Tissues were isolated from indicated C57BL/6 or iNKT TN mouse lines and stained with anti-CD3 and CD1d-(PBS57)-tetramer.

C   Spleen, mLN, sdLN, liver, adipose, and lung lymphocytes from Jα18$^{-/-}$ or Vα14 mice were stimulated *in vitro* with RAW-CD1d cells and 1 μg α-GalCer. An additional sample of Vα14 lymphocytes from each organ was plated with RAW-CD1d cells but no α-GalCer. Supernatants were collected after 24 h and cytokine concentration determined by cytokine bead array. Error bars are SD of mean values from three different mice per group. Results shown are representative of two independent experiments where *n* = 3 biological replicates.

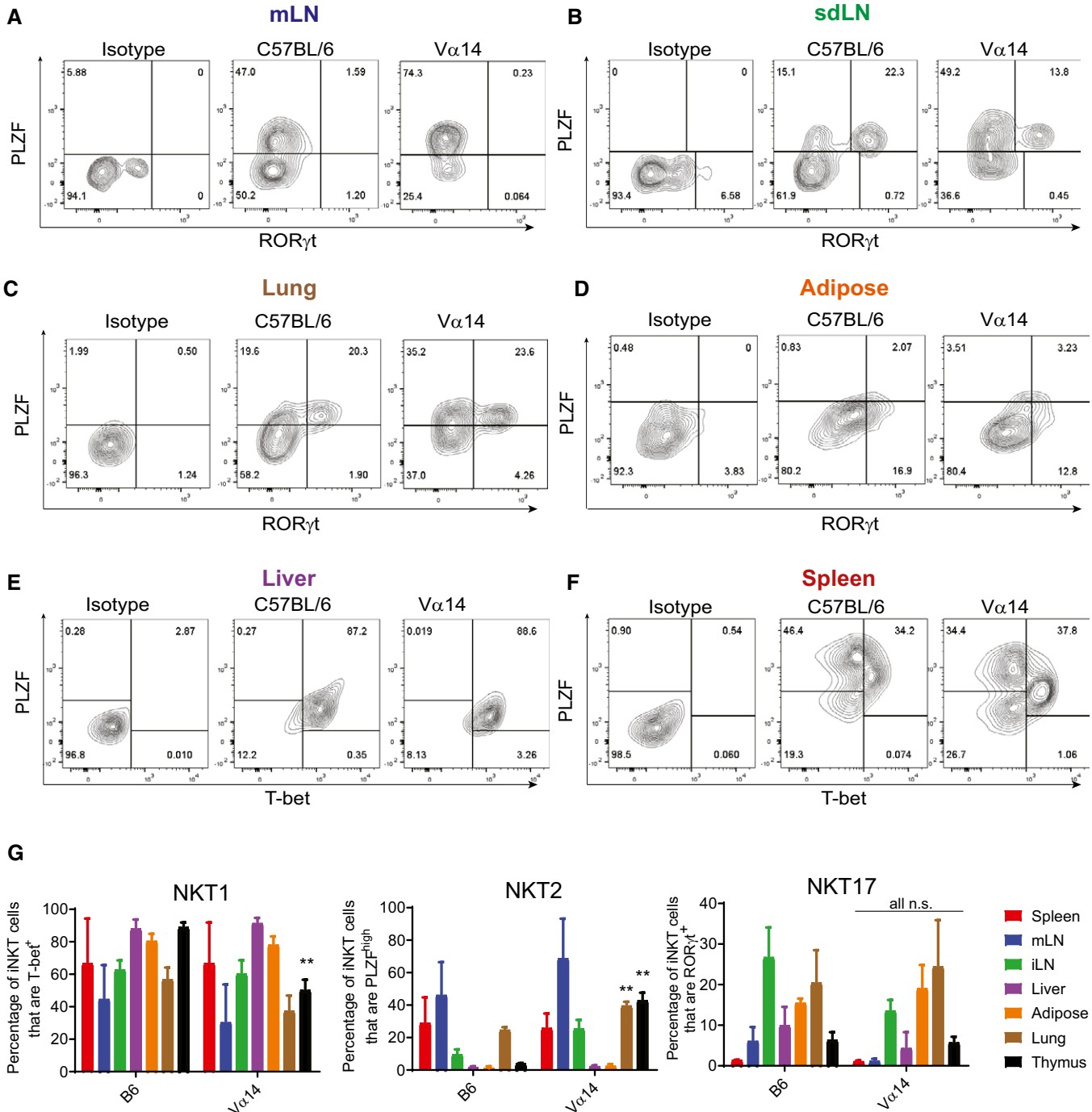

**Figure 2.  iNKT cells from TN and C57BL/6 mice show similar influence of tissue microenvironment on NKT1, NKT2, and NKT17 subsets.**

A–F   Lymphocytes from the indicated tissues of C57BL/6 and Vα14 mice were stained with anti-CD3 and CD1d-(PBS57)-tetramer, before they were fixed, permeabilized, and stained with antibodies to T-bet, RORγt, and PLZF. Results shown are gated on CD3+CD1d-tetramer+ cells.

G     The percentage of CD3+ CD1d-tetramer+ iNKT cells in each organ that stained positively for PLZF, T-bet, and RORγt are shown. **$P < 0.01$, Mann–Whitney test. Error bars are SD.

Data information: Results shown are representative of three independent experiments where $n = 3$ biological replicates.

Thompson *et al*, 2017). However, across all peripheral tissues, Vα14 TN iNKT cells showed similar frequencies of NKT1/2/17 cells when compared to iNKT cells in those same tissues from C57BL/6 mice, with the exception of slightly higher frequencies of NKT2 cells in the lungs of Vα14 TN mice (Fig 2G). Inguinal LN iNKT cells from both Vα14 and C57BL/6 mice had an increased frequency of RORγt+ NKT17 cells—a population that was notably absent from mesenteric LN iNKT cells in both groups of mice. Liver iNKT cells

were more strongly T-bet$^+$, indicating an increased frequency of NKT1 cells in the liver (Fig 2G). Adipose iNKT cells are among the more distinct iNKT cell lineages and express the transcription factor E4BP4 rather than PLZF. We analyzed adipose iNKT cells from Vα14 TN mice and found low levels of PLZF and high expression of E4BP4, similar to C57BL/6 mice and previous reports (Fig EV2A–D and Lynch *et al*, 2015).

### iNKT cells are found in Peyer's patches of both wild-type and Vα14 TN mice and correlate with increased IgG1$^+$ B cells

iNKT cells with follicular helper-like function have been previously defined; immunization with α-GalCer induces the formation of NKT$_{fh}$ in the spleen (Chang *et al*, 2011). However, NKT$_{fh}$ have not previously been shown in Peyer's patches, an important site for germinal center formation that is continuously exposed to gut antigens. When we examined Peyer's patches from Vα14 TN mice, we found CD1d-tetramer$^+$ iNKT cells at greatly increased frequency compared to C57BL/6 mice (Fig 3A and B). Importantly, we could detect a small population of iNKT cells in C57BL/6 mice as compared to Jα18$^{-/-}$ mice, indicating that iNKT cells are also present in Peyer's patches in wild-type mice (Fig 3A and B).

Peyer's patches are important sites of germinal center activity to produce antigen-specific antibodies (Reboldi and Cyster, 2016). We first measured fecal IgA titers and found only modest differences in total IgA among mice with low, high, and zero levels of PP-NKT cells (Fig 3C). IgA can be produced in both T-cell-dependent and T-cell-independent fashion and correlates with the amount of TGF-β present in Peyer's patches. Although IgA is the predominant isotype in the gut lumen, IgG is also secreted into and recycled from the gut lumen through binding to FcRn (Rath *et al*, 2013). IgG sampling of gut luminal contents is an important source of antigen acquisition and has a protective role against some enteric pathogens (Bry & Brenner, 2004; Maaser *et al*, 2004). To investigate IgG antibody production, we measured IgG1 titers by ELISA and IgG1$^+$ B cells by flow cytometry (Fig 3D–F). IgG1$^+$ B-cell frequencies were similarly low (< 1%) in spleens of C57BL/6, Vα14 TN, and Jα18$^{-/-}$ mice; total serum IgG1 was also not different among the groups. However, C57BL/6 and Vα14 mice had significantly increased numbers of IgG1$^+$ B cells in both mLN and Peyer's patches and compared to Jα18$^{-/-}$ mice (Fig 3F), and increased fecal IgG1 titers (Fig 3D). Jα18$^{-/-}$ mice have somewhat limited TCR repertoire diversity (Bedel *et al*, 2012); thus, we also examined fecal IgG1 in CD1d$^{-/-}$ mice and age- and sex-matched C57BL/6 control mice. Both fecal IgG1 and the frequencies of IgG1$^+$ B cells in mLN and Peyer's patches were reduced in CD1d$^{-/-}$ mice (Fig EV3A and B). Thus, PP-NKT cells are critical for homeostatic levels of IgG1$^+$ B cells in the gut, but do not appear to be dose-limiting, as even the low frequencies of iNKT cells found in wild-type mice are sufficient to allow for class switching to IgG1. This requirement of iNKT cells for IgG1$^+$ B cells in Peyer's patches was observed across two different mouse facilities (Fig EV3C).

### PP-NKT cells provide indirect help to B cells through production of IL-4 and are important in oral vaccination

To determine how PP-NKT cells might regulate B-cell class switching, we sorted PP-NKT cells from Vα14 TN, as well as iNKT cells

from spleen and CD4 T cells from Peyer's patches. Cell yields were adequate such that transcriptional profiling could be performed on bulk populations of cells isolated from 3 individual mice ($n = 3$ biological replicates). Cluster analysis revealed that PP-NKT cells were more similar to spleen iNKT cells than CD4 T cells, thereby confirming their identity as *bona fide* iNKT cells (Fig 4A). Genes associated with Tfh cell identity or required for their function were highly expressed in Peyer's patch CD4 T cells, but absent from PP-NKT (Fig 4B). Notably, PP-NKT expressed undetectable levels of CD40L and CXCR5, making it unlikely that PP-NKT cells make direct cell–cell contact with germinal center B cells.

IL-4 is important for regulating B-cell class switching to IgG1, and early production of IL-4 by iNKT cells in the lung was previously reported to be critical for supporting B cells *en route* to germinal centers (Gaya *et al*, 2018). We therefore examined IL-4 and IFNγ production from cells cultured from the spleen, mLN, or Peyer's patches (Fig 4C and D). Following stimulation, we analyzed the relative proportions of iNKT cells and non-iNKT T cells producing IL-4 or IFNγ and found that the majority of the IL-4 was derived from iNKT cells. In contrast, although iNKT cells produced IFNγ, the majority of the IFNγ was derived from other T cells in the cultures. We therefore conclude that PP-NKT cells could be an important local source of IL-4, which supports B-cell class switching and production of IgG1. To determine the relevant function of PP-NKT cells *in vivo*, we challenged mice with α-GalCer either intravenously or by oral gavage. Mice were treated with brefeldin A to prevent cytokine secretion, and then, cells from the indicated tissues were analyzed by intracellular cytokine staining (Fig 4E and F). NKT cells from spleen produced both IFNγ and IL-4 upon intravenous α-GalCer, but oral gavage failed to induce activation of spleen iNKT (Fig 4E and F). Peyer's patch iNKT cells produced primarily IL-4 upon oral administration of α-GalCer.

Vα14 TN iNKT cells pooled from spleen and LNs, and cocultured with naïve B cells and α-GalCer yielded no detectable IgG1, consistent with the lack of detectable CD40L expression by iNKT cells (Figs 4B and 5A). Provision of agonistic anti-CD40 induced robust B-cell activation as evidenced by IgM secretion. IgM levels were not augmented by the presence of iNKT cells (Fig 5B). In contrast, IgG1 production was dependent on the presence of iNKT cells, was increased by α-GalCer, and was diminished by the addition of blocking antibodies to CD1d that prevented iNKT cell activation in this setting (Fig 5B). To determine whether iNKT cell recognition of CD1d on B cells was important for induction of IgG1, we modeled the iNKT-B cell interaction *in vitro* using iNKT cells obtained from skin-draining LN, mesenteric LN, or Peyer's patches of Vα14 TN mice or from Peyer's patches of IL-4$^{-/-}$ mice (Fig 5C–F). These iNKT cells were cocultured with CD40-activated B cells obtained from wild-type or CD1d$^{-/-}$ mice. Vα14 TN iNKT cells from all three tissues produced IL-4, with mLN and PP-iNKT cells producing more IL-4 than sdLN (Fig 5C). IL-4 was not detected from IL-4$^{-/-}$ PP cells. IgG1$^+$ class-switched B cells and IgG1-secreted Ab were strongly induced in cocultures of B cells with Vα14 TN mLN and PP-iNKT cells, and this induction was blocked by addition of blocking antibodies to IL-4 (Fig 5D–F). CD1d$^{-/-}$ B-cell cocultures phenocopied WT B-cell cocultures, indicating that direct recognition of CD1d on B cells is not required (Figs 5D–G and EV3D). Rather, IL-4 produced by iNKT cells induced B-cell class switching to IgG1 *in vitro*, and we propose

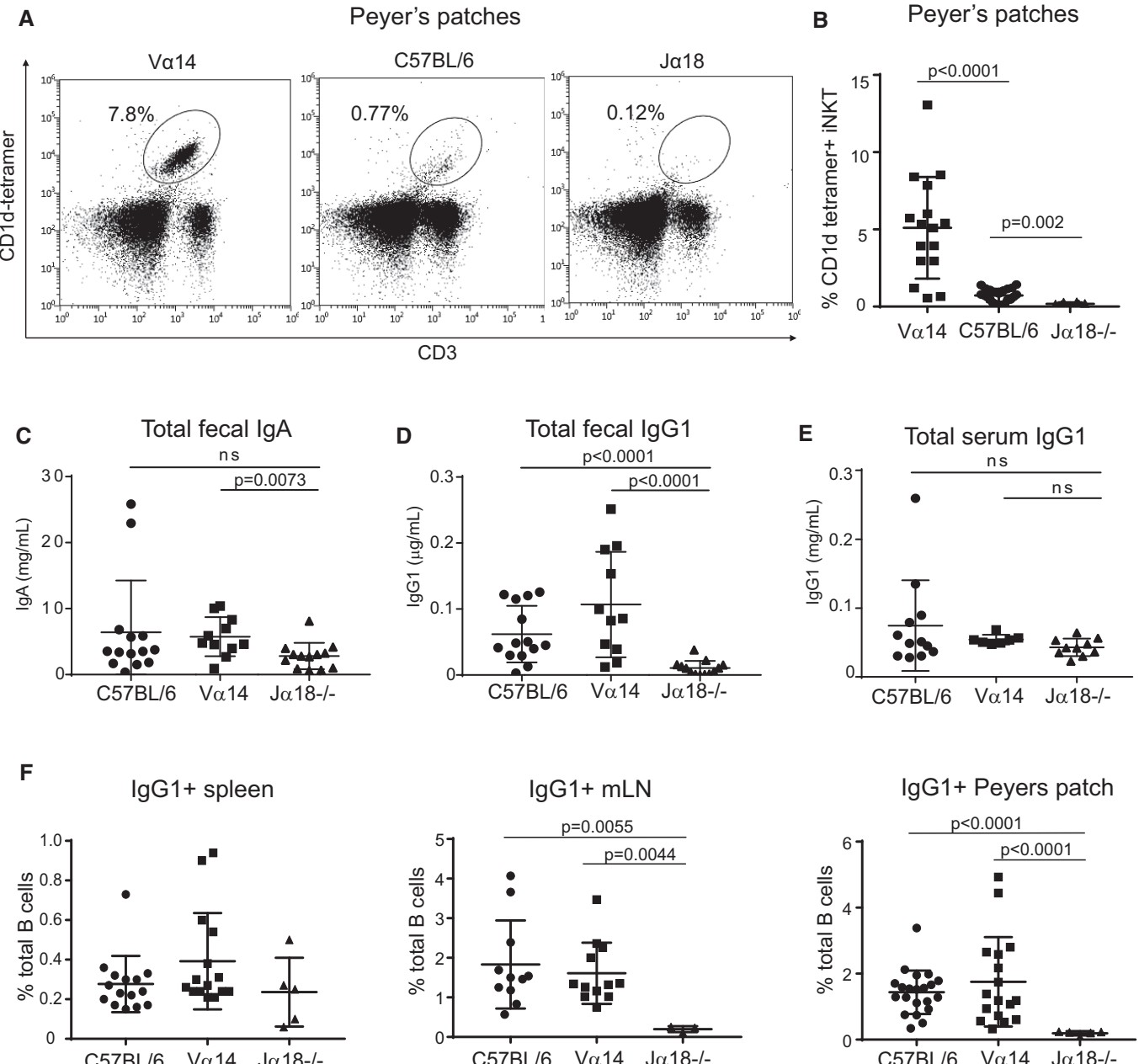

**Figure 3.   iNKT TN mice show increased IgG1 production and IgG1+ B cells in the mLN and Peyer's patches.**

A   Lymphocytes from Peyer's patches of C57BL/6 and Vα14 mice were stained with anti-CD3 and CD1d-(PBS57)-tetramer.

B   Percentage of lymphocytes that were CD3+CD1d-tetramer+ iNKT cells among Peyer's patches of C57BL/6, Vα14, and Jα18−/− mice are shown. Mann–Whitney test. Error bars are SD. C57BL/6 *n* = 22; Vα14 *n* = 16; Jα18−/− *n* = 8.

C–E   Mice were analyzed for total fecal IgA (C), total fecal IgG1 (D), or total serum IgG1 (E) by ELISA. Mann–Whitney test. Error bars are SD. C57BL/6 *n* = 14; Vα14 *n* = 11; Jα18−/− *n* = 13.

F   Percentages of total B cells that were IgG1+ in the spleen, mLN, and Peyer's patches of C57BL/6, Vα14, and Jα18−/− mice are shown. Mann–Whitney test. Error bars are SD. C57BL/6 *n* = 15; Vα14 *n* = 15; Jα18−/− *n* = 5.

this as a likely mechanism by which iNKT cells in Peyer's patches may be inducing IgG1 *in vivo*.

The iNKT cell agonist lipid α-GalCer has been proposed as a potential adjuvant for both preventative vaccines against pathogens and therapeutic cancer vaccines (Silk *et al*, 2004; Singh *et al*, 2014;

Tefit *et al*, 2014; Kharkwal *et al*, 2016; Khan *et al*, 2017; Li *et al*, 2017; Wolf *et al*, 2018). Given the location of PP-NKT and their positioning as potential first responders, we speculated that α-GalCer could be used as an adjuvant for oral protein-based vaccines. Oral vaccination of mice with the model antigen ovalbumin

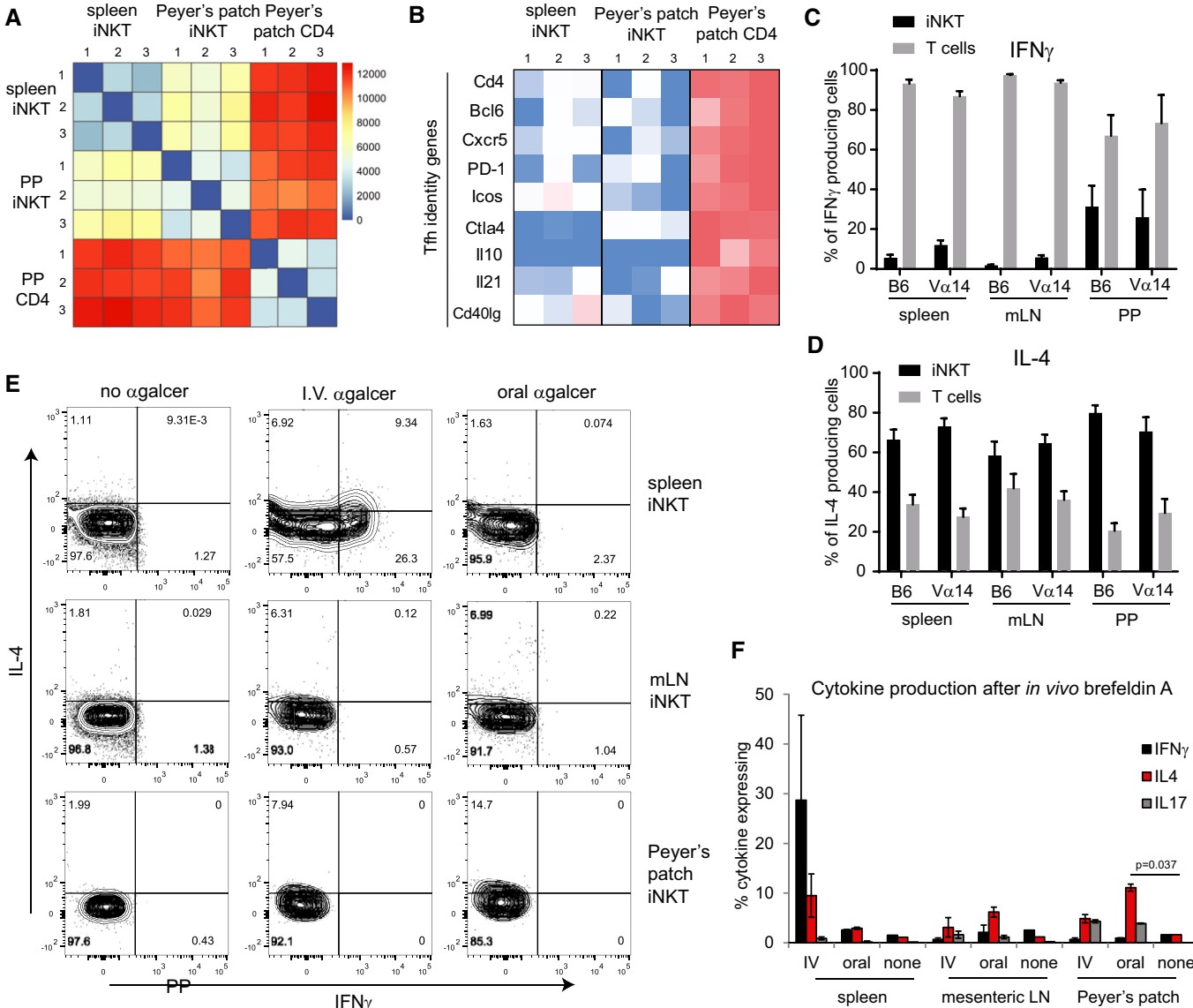

**Figure 4. PP-NKT cells produce IL-4 *in vitro* and *in vivo*.**

A   CD1d-(PBS57)-tetramer⁺ CD3⁺ cells were sorted from spleens or PP of 3 different Vα14 TN mice along with CD4⁺CD3⁺CD1d-tetramer⁻ cells from PP (PP CD4). RNAseq was performed.

B   Heatmap of FPKM values for the indicated Tfh genes across each RNAseq sample.

C, D   Spleen, mLN, and PP lymphocytes from Vα14 and C57BL/6 mice were stimulated with PMA and ionomycin. Lymphocytes were stained with anti-CD3, CD1d-(PBS57)-tetramer, anti-IL-4, and anti-IFNγ. Percentages of iNKT cells and non-iNKT T cells within the population of CD3⁺IL-4⁺ cells and CD3⁺IFNγ⁺ cells, *n* = 5 mice for C57BL/6 group and *n* = 4 mice for Vα14 group. Error bars are SD.

E   Vα14 TN mice were administered α-GalCer either 2 µg intravenously or 5 µg by oral gavage. Mice were given brefeldin A intraperitoneally after 30 min, and tissues were harvested 3 h later. Cells from spleen, mesenteric lymph node, and Peyer's patches of Vα14 TN mice were permeabilized, fixed, and stained with antibodies to IL-4, IFNγ, and IL-17. Plots shown are gated on CD1d-(PBS57)-tetramer⁺ CD3⁺ iNKT cells.

F   Quantification of data from (E), *n* = 2 mice per group. Mann–Whitney test. Error bars are SEM.

produced no appreciable IgA- or IgG1-specific antibody responses in serum or feces, even after multiple boosts (Figs 6A–C, and EV4A and B). Addition of α-GalCer admixed into the oral vaccine generated robust serum IgA and IgG1 responses in half of the mice after one boost (Day 15, Fig EV4A), and fecal and serum anti-OVA responses in all mice after multiple doses (Figs 6B and EV4A–C). Interestingly, Vα14 TN mice did not show augmented antibody

titers, suggesting that even small numbers of PP-NKT cells are adequate to serve as a mucosal adjuvant.

To determine whether PP-NKT cells are related to the well-described NKT1, NKT2, and NKT17 subsets, we stained freshly isolated lymphocyte preparations from spleen, mesenteric lymph nodes, or Peyer's patches with antibodies to the transcription factors T-bet, PLZF, and RORγt. By this analysis, PP-NKT contained an

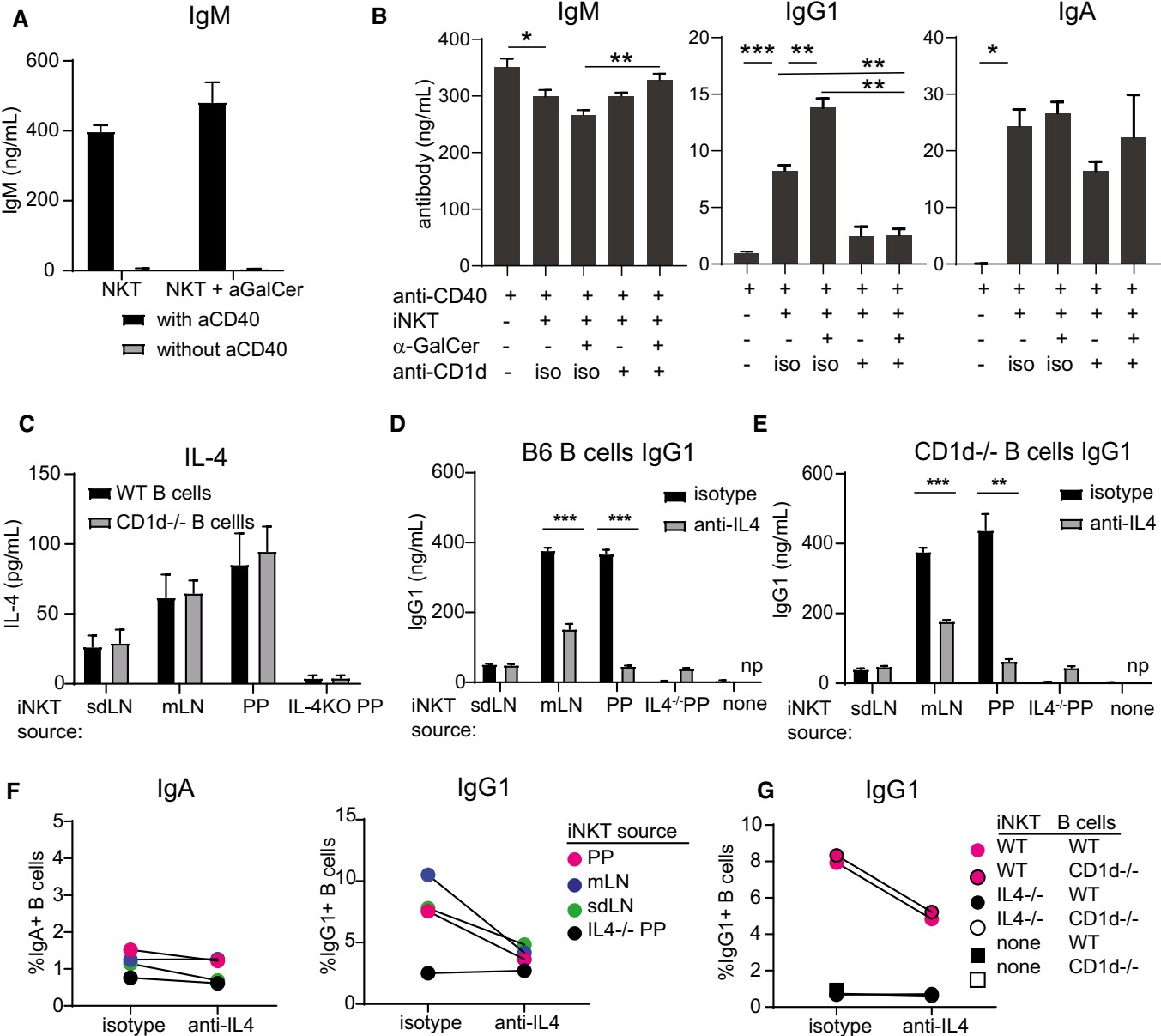

**Figure 5. iNKT cells provide indirect help for B-cell class switching to IgG1 *in vitro*.**

A   Pooled spleen and LN cells from a Vα14 TN mouse were cocultured with wild-type B cells with or without 1 μg α-GalCer and with or without agnostic anti-CD40. IgM was measured by ELISA of culture supernatants 4 days later.

B   Pooled spleen and LN cells from a Vα14 TN mouse were cocultured with anti-CD40-activated wild-type B cells with or without 1 μg α-GalCer and with blocking antibody to CD1d (1 μg/ml, clone 1B1) or isotype control as indicated. IgM, IgG1, and IgA were measured by ELISA of 4-day culture supernatants.

C–G   Spleen, mLN, or PP cells from Vα14 TN iNKT mice and PP cells from an IL-4$^{-/-}$ mouse were cocultured with anti-CD40-activated wild-type or CD1d$^{-/-}$ B cells. 1 μg α-GalCer was added to the cultures to specifically activate iNKT cells. Blocking antibodies to IL-4 or isotype were added as indicated. IL-4 (C) and IgG1 (D, E) were measured by ELISA of culture supernatants. IgG1$^+$ and IgA$^+$ class-switched B cells were enumerated by flow cytometry after 4 days of coculture (F, G). Representative of two independent experiments. np = not performed.

Data information: Mann–Whitney test. Error bars are SD of triplicate samples. *$P < 0.05$; **$P < 0.01$; ***$P < 0.001$.

unusually high fraction of T-bet$^+$ cells, suggesting that these might be related to NKT1 cells (Fig 6D and E). However, oral administration of α-GalCer did not result in IFNγ production from PP-NKT cells, demonstrating that although PLZF$^{high}$ cells appear to be a minority of the population, PP-NKT cells produce IL-4, not IFNγ, in the setting of oral vaccination (Figs 4F and 6E).

**PP-NKT cells exhibit a unique gene expression profile**

NKT subsets express unique gene signatures, including the canonical transcriptional factors and cytokines, as well as a profile of other differentially expressed genes. We used previously published gene lists that had been identified from single-cell transcriptional profiling

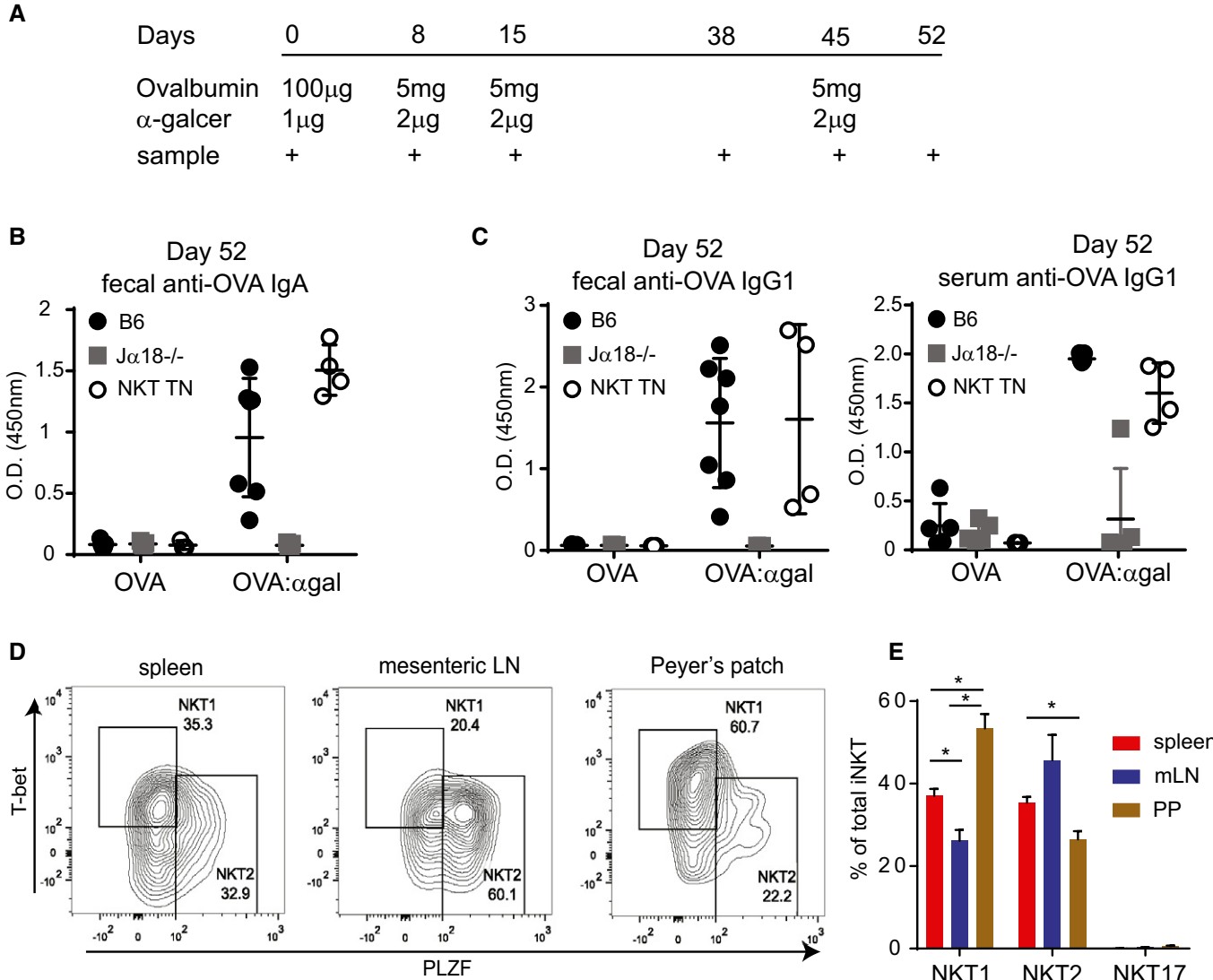

**Figure 6. Oral vaccination with α-GalCer leads to specific antibody production in NKT cell proficient mice.**

A   Mice were administered ovalbumin by oral gavage, with or without α-GalCer at the indicated time points, at the indicated amounts. Mice used were Jα18[−/−] (n = 10 total), iNKT transnuclear (n = 7 total), or littermate controls of iNKT TN mice that did not inherit the Vα14 allele (B6, n = 12 total).

B, C   OVA-specific antibody titers from feces and blood were determined by ELISA from the mice in (A). Shown are optical density values for samples collected on Day 52 post-vaccination (7 days after the final boost). Error bars are SD.

D   Cells from spleen, mesenteric lymph node, and Peyer's patches of Vα14 TN mice were permeabilized, fixed, and stained with the indicated antibodies. Plots shown are gated on CD1d-(PBS57)-tetramer[+] CD3[+] iNKT cells.

E   Quantification of data from (D), n = 3 mice per group. Mann–Whitney test. Error bars are SEM. *P < 0.05.

of iNKT cell subsets (Engel *et al*, 2016). We compared expression levels of these genes between spleen and PP-NKT, and found that PP-NKT cells do not upregulate the canonical transcriptional signatures associated with NKT1, NKT2, NKT17, and NKT0 cells, or at least that no NKT subset is enriched compared to spleen (Appendix Fig S1). This suggests that PP-NKT cells constitute a novel NKT cell population with strong tissue-specific imprinting, such as adipose iNKT cells, or that they are comprised of a highly heterogeneous population of NKT cells, such as the splenic iNKT cell pool. To determine whether iNKT cells could seed the PP in adult mice, we adoptively transferred Vα14 TN cells into sublethally

irradiated Jα18[−/−] hosts. Transferred iNKT cells could be detected in spleen, liver, adipose tissue, and lymph nodes; however, we did not detect transferred iNKT cells in Peyer's patches of irradiated Jα18[−/−] mice post-transfer (Fig EV5A and B).

Although PP-NKT cells bear many similarities to iNKT cells from spleen, principal component analysis reveals that PP-NKT cells form a distinct cluster (Fig 7A). Differential gene expression analysis comparing spleen iNKT with PP-NKT shows strong upregulation of the gut-homing chemokine receptor CCR9, as well as hallmarks of tissue residency (CD69 and CD103; Fig 7B). This signature of tissue residency was uniquely expressed in PP-NKT cells, along with

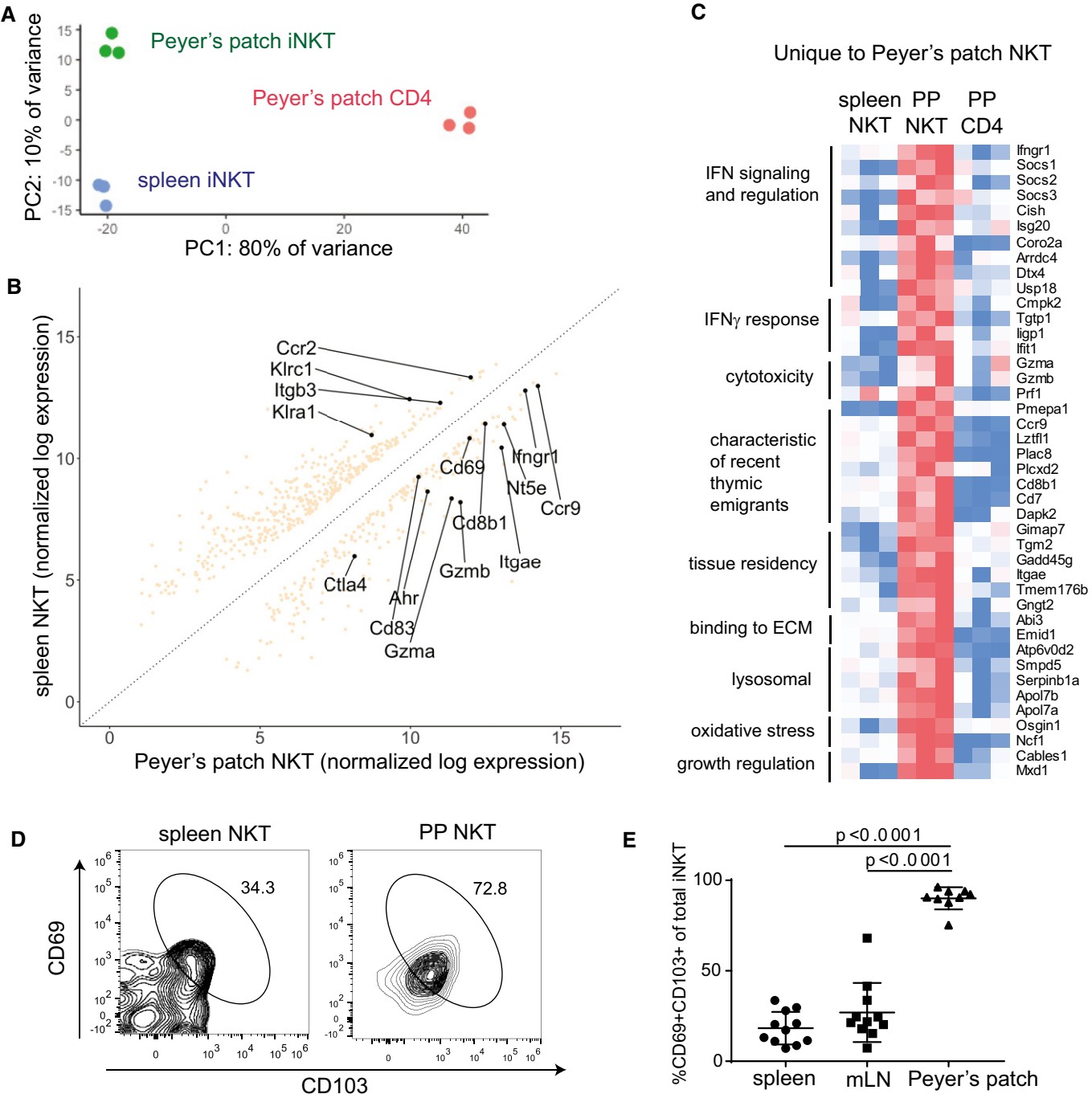

**Figure 7. PP-iNKT cells exhibit a unique gene expression profile compared to spleen iNKT and PP CD4+ cells.**

A  Principal component analysis of RNAseq samples from biological replicates of PP-iNKT, spleen iNKT, and PP CD4+ cells

B  Differential expression analysis comparing transcriptomes of spleen iNKT cells versus PP-iNKT cells.

C  Heatmap of FPKM values from genes uniquely upregulated in PP-iNKT cells as compared to both spleen iNKT and PP CD4 cells. Red indicates higher expression, and blue indicates lower expression.

D  Spleen and PP cells from Vα14 TN mice were stained with the indicated antibodies, analyzed by flow cytometry, and gated on CD1d-(PBS57)-tetramer+ CD3+ cells.

E  Quantification of data from (D), n = 11 mice per group. Mann–Whitney test. Error bars are SEM.

several genes characteristic of recent thymic emigrants (Fig 7C). CD103 and CD69 expression on PP-NKT were confirmed by flow cytometry and suggest that PP-NKT cells are tissue-resident cells

(Fig 7C and D), similar to a previous report showing an increased tissue residency signature in lung iNKT cells compared to iNKT cells from spleen (Salou *et al*, 2019). PP-NKT cells were uniquely

responsive to type I and type II IFNs as shown by expression of IFN response genes (Fig 7C). PP-NKT cells share several features with CD4$^+$ Tregs (expression of both CTLA-4 and the adenosine-converting enzyme CD73), suggesting possible regulatory function or maintenance of mucosal tolerance at homeostasis. Aryl hydrocarbon receptor, which has been implicated in sensing colitis-inducing oxazoles (Iyer *et al*, 2018), is upregulated in PP-NKT cells. PP-NKT cells also uniquely express granzymes A and B, along with perforin, suggesting cytotoxic potential (Fig 7C).

## Discussion

We here report PP-NKT as a novel population of iNKT cells with unique function in supporting IgG1 class switching in the gut mucosa. Gut-resident B cells primarily produce IgA or IgG1 for secretion into the gut lumen. IgA is by far the most abundant secreted isotype, and class switching to IgA can occur independently of T-cell help and is correlated with the amount of TGF-β present. Class switching to IgG1 is T-cell-dependent and occurs in Peyer's patches and mesenteric lymph nodes, both of which contain significant populations of CD4 Tfh cells. Our finding of iNKT cells in Peyer's patches suggests that these cells may play a role in B-cell class switching, and indeed, we show that PP-NKT cells express the majority of the IL-4 produced in these tissues, even when PP-NKT cells are present at low frequencies such as in C57BL/6 mice.

PP-NKT cells are rare. Their presence in wild-type mice was only appreciated by first finding them in Vα14 TN mice, thus highlighting the utility of the transnuclear approach. Indeed, all tissue-resident iNKT cell populations are many-fold more abundant in our panel of iNKT TN mice. We show by cytokine production and transcription factor expression that tissue-resident iNKT cells from lungs, adipose, liver, mesenteric lymph nodes, and skin-draining lymph nodes faithfully recapitulate the properties of wild-type iNKT cells from those same tissues. Comparison of monoclonal iNKT cells using Vβ7 or Vβ8 TCRs with different preferences for CD1d–lipid complexes showed little influence of the iNKT TCR on development of tissue-resident iNKT cells, although the monoclonal TN NKT mouse lines may be useful for studying responsiveness to particular lipids in different tissues.

PP-NKT cells are a clearly distinct population with a unique gene expression signature, suggesting a long-lived tissue niche. Our attempts to adoptively transfer iNKT cells into naïve Jα18$^{-/-}$ mice resulted in undetectable recovery of iNKT cells in Peyer's patches. We propose that these cells seed Peyer's patches early in the postnatal period and are non-recirculating, as was previously reported for total gut iNKT cells (Olszak *et al*, 2012; An *et al*, 2014). There, they likely contribute to mucosal inflammation and tolerance by a variety of mechanisms. We report one effect of early production of IL-4 in promoting B-cell class switching to IgG1, a property that can be exploited for oral vaccination. However, this is unlikely to be the only function of PP-NKT cells, and we predict that multiple avenues for regulation by PP-NKT cells will be uncovered in the future. Indeed, several commensal-derived lipids have been reported to regulate gut-resident iNKT cells (Olszak *et al*, 2012; Wingender *et al*, 2012; An *et al*, 2014), and conversely, the composition of the microbiome appears to be regulated by CD1d$^+$ cells and iNKT cells (Nieuwenhuis *et al*, 2009; Selvanantham *et al*, 2016; Saez de

Guinoa *et al*, 2018). Intriguingly, when we analyzed IgG1$^+$ B cells from iNKT TN mice versus their littermate controls housed in two different mouse facilities, we found that in one mouse facility, the frequencies of IgG1$^+$ B cells were increased in iNKT TN compared to littermates, and in the second mouse facility, IgG1$^+$ B cells were similar between iNKT TN and littermate controls. In the interest of not overstating our claims, we have chosen to report results from the second mouse facility, where the differences between iNKT TN mice and littermate controls were less pronounced. In both facilities, Jα18$^{-/-}$ mice had fewer IgG1$^+$ B cells, thus supporting our conclusion that PP-NKT cells are required for homeostatic levels of gut IgG1. However, the magnitude of the PP-NKT cell effect changed between mouse facilities, thereby suggesting a complex link with the microbiome that is worth further exploration.

PP-NKT cells produce IL-4 in the absence of IFNγ, thereby providing a mechanism for the observed increase in IgG1$^+$ Peyer's patch B cells and fecal antibodies. How they produce IL-4 is unclear. NKT2 cells are under-represented in PP-NKT compared to spleen or mesenteric lymph nodes. We observed IL-4 production from 11% of total PP-NKT cells after oral α-GalCer, so preferential activation of NKT2 cells is formally possible, for example, by NKT2 cells being better positioned proximal to CD1d$^+$ antigen-presenting cells. This seems unlikely since administration of α-GalCer by either intravenous or oral routes both resulted in IL-4 production without IFNγ; however, further understanding of the spatial orientation of the relevant CD1d$^+$ cells in Peyer's patches would be necessary to answer this question. Another possibility is that PP-NKT cells are indeed NKT1 cells that would co-express IFNγ and IL-4, but that IFNγ is specifically suppressed in Peyer's patch setting, possibly due to increased adenosine levels (Lappas *et al*, 2005). In either case, the production of IL-4 is important for homeostatic levels of fecal IgG1 and can also be exploited for oral vaccination. Since oral delivery of α-GalCer stimulated IL-4 production from PP-NKT, but had minimal effect on iNKT cells in the spleen, the effects of oral α-GalCer may be safer than systemic administration. Humans have iNKT cell frequencies that are overall lower and more variable than those seen in mice. We discovered PP-NKT by looking in our transnuclear mice, and PP-NKT cells exist at small frequencies in wild-type mice. However, even rare populations of iNKT cells can produce copious IL-4, and we have shown that the very few iNKT cells found in wild-type mouse are capable of supporting steady-state IgG1 and specific antibody production in the setting of oral vaccination. These data suggest that a minimum threshold number of PP-NKT cells are adequate for IgG1 class switching, but that increased iNKT cells are not necessarily correlated with higher IgG1 titers. PP-NKT may be useful for both routine surveillance of mucosal pathogens, and as a cellular adjuvant for oral vaccines.

## Materials and Methods

### Animal care

Animals were housed at the Dana-Farber Cancer Institute and were maintained according to protocols approved by the DFCI IACUC. C57BL/6, CD1d$^{-/-}$, and IL-4$^{-/-}$ mice were purchased from Jackson Labs. Jα18$^{-/-}$ mice were obtained from Dr. Michael Brenner (Boston, MA). Vα14, Vβ7A, Vβ7C, and Vβ8 iNKT TN mouse lines were

generated using somatic cell nuclear transfer as previously described (Dougan et al, 2012, 2013a,b).

## Tissue preparation

Spleen, thymus, Peyer's patches, and lymph nodes were harvested and homogenized through a 40-μm cell strainer. Adipose tissue was minced with scalpels prior to digestion with 5 mg/ml collagenase II for 30 min while rotating at 37°C. Lung tissue was placed in a gentleMACS C Tube (Miltenyi 130-093-237) and digested using the lung dissociation kit enzymes (Miltenyi 130-095-927) and the gentle-MACS Dissociator (Miltenyi 130-093-235), as per the manufacturer's recommendation. Liver was homogenized through a 70-μm cell strainer and centrifuged at 300 g for 5 min. The organ pellet was resuspended in 10 ml of 35% Percoll (GE Healthcare 17-0891-01) in RPMI. 5 ml of 70% Percoll in PBS was subsequently added to form a bottom layer in the tube before centrifugation at 450 g for 15 min with no brakes. After centrifugation, the middle layer of lymphocytes was harvested into 10 ml PBS.

## Flow cytometry

Cell preparations from spleen, thymus, lymph nodes, liver, epididymal fat pads, lung, or Peyer's patches were harvested and exposed to hypotonic lysis to erythrocytes. Following cell preparation, cells were stained and analyzed using a BD LSRFortessa and a Sony Spectral Flow Cytometer. CD1d-PBS57 (CD1d-αgal) tetramers were obtained from the NIH Tetramer Core Facility. The following antibodies used for staining were obtained from BioLegend: IFNγ (Clone XMG1.2, Cat 505830), IL-4 (Clone 11B11, Cat 504109), T-bet (Clone 4B10, Cat 644816), CD3ε (Clone 17A2, Cat 100241), GL7 (Clone GL7, Cat 144609), B220 (Clone RA3-6B2, Cat 103243), IgG1 (Clone RMG1-1, Cat 406610), IgG2b (Clone RMG2b-1, Cat 406707), and IgD (Clone 11-26c.2a, Cat 405711). The following antibodies were from eBioscience: RORγt (Clone B2D, Cat 17-6981-80) and PLZF (Clone Mags.21F7, Cat 53-9320-82). The following antibody is from BD Pharmingen: IgA (Clone C10-3, Cat 559354).

## Stool sample generation

Individual stool samples from C57BL/6 and Vα14 mice were collected and normalized to their weight by adding volumes of distilled water proportional to their weight (1 g stool:10 ml H$_2$O). Samples were vortexed to mix and incubated at 37°C for 15 min to loosen the stool. Samples were vortexed again and centrifuged at 450 g for 1 min. For some experiments, fecal samples were centrifuged at 16,000 g for 5 min to pellet bacteria. Supernatant was collected into a new tube and frozen at −20°C until use. Negligible differences in antibody titers were observed between the same samples centrifuged at low speed versus high speed.

## ELISA

High-binding assay plates (Corning 9018) were coated with anti-Ig (H + L) antibody (Southern Biotech 103101) at a 1:500 dilution in PBS or with ovalbumin (Sigma-Aldrich, 100 ng/ml). Plates were allowed to coat overnight at 4°C. Plates were subsequently washed with wash buffer (1:2,000 dilution of Tween in PBS) and blocked

with assay diluent (10% FBS in PBS) for 1 h at room temperature (RT). Plates were washed again with wash buffer before addition of 100 μl of samples, diluted at 1:2 and 1:10 for IgG1 and IgA. Plates were incubated overnight at 4°C and then washed with wash buffer. 100 μl of secondary antibody (1:5,000 in assay diluent) was added to all wells [IgA (Southern Biotech 1040-05) and IgG1 (Southern Biotech 1071-05)] and incubated for 1 h at RT. After washing with wash buffer, 100 μl of tetramethylbenzidine (Sigma-T8665) was added and incubated for 10–20 min at RT; then, 50 μl of 1 M hydrochloric acid was added to stop the reaction. Optical density values were read as a measure of concentration at 450 nm on a plate reader.

## Cell culture

Cells were cultured in RPMI 1640 medium supplemented with 10% heat-inactivated FBS, 100 U/ml penicillin G sodium, 2 mM L-glutamine, 1 mM sodium pyruvate, 100 μg/ml streptomycin sulfate, 0.1 mM non-essential amino acids, and 0.1 mM 2-ME. RAWd cells (gift from Dr. Michael Brenner) were cultured in DMEM with 10% FBS, 1% PenStrep, and 2 mM L-glutamine. CD1d expression on RAWd cells was confirmed by inclusion of a no α-GalCer condition in each experiment. RAWd cells were tested for mycoplasma every 4 months. For cocultures, total cell preparations from the indicated organs were added to RAWd cells pulsed with 1 μg/ml α-GalCer (Avanti Lipids). RAWd cells were plated into flat-bottom 96-well plates at 50,000 cells per well. 1/60 of spleen was added to culture. For lymph nodes, liver, and adipose tissue, 1/3 of the organ was added per well. The contents of one lung lobe (mouse left) were harvested, and 1/3 of the organ was added per well. Production of IL-4, IL-2, GM-CSF, IL-17, IFNγ, and IL-10 of 24-h culture supernatants was measured by ELISA, as indicated (BioLegend). 31-plex cytokine and chemokine panel bead array analysis was performed by Eve Technologies. For stimulation experiments, cell cultures were stimulated with PMA and ionomycin for 4 h, with the addition of GolgiStop (Invitrogen). Cells were subsequently fixed, permeabilized, and stained with Abs to IFNγ and IL-4.

## Oral vaccinations

Ten- to 18-week-old mice of both sexes were used for vaccination. Wild-type mice used were cohoused with littermates of the transnuclear NKT cell mice. Jα18$^{−/−}$ mice were bred separately. Mice were given by oral gavage 5 mg ovalbumin suspended in 150 μl of sterile water with or without 2 μg α-galactosylceramide (Avanti Lipids). Mice were immunized and boosted according to the schedule shown in Fig 6A.

## In vivo cytokine analysis

Ten- to 18-week-old Va14 iNKT TN mice were given by oral gavage 5 μg α-galactosylceramide (Avanti Lipids) in 150 μl sterile water or 2 μg α-galactosylceramide in 150 μl PBS intravenously. Mice were housed in standard caging for 30 min, then injected intraperitoneally with 5 μg brefeldin A in 150 μl PBS, and returned to standard caging for an additional 3 h. Tissue lymphocytes were harvested, fixed and permeabilized, and stained with antibodies to IL-4, IFNγ, and IL-17.

## RNA sequencing

Peyer's patch cells and spleen cells were prepared from three littermate Va14 female mice. Cell preparations were stained with CD1d-PBS57 tetramer and antibodies to CD4, CD45, and CD3. iNKT cells were sorted by FACS from both tissues and CD4$^+$ CD1dtet- cells were sorted from Peyer's patches into collection tubes containing RNA isolation buffer (Qiagen RNA mini plus). RNA was prepared as per the manufacturer's protocol. Library construction and Illumina sequencing were performed by the DFCI Molecular Genomics Core Facility. Raw transcript counts were collapsed to gene-level counts and log-normalized using the R package DESeq2. Principal component analysis was then performed on the normalized counts. Principal component analysis was performed using R. Total variance was 1128.203.

## Statistics

Error bars are SD unless otherwise noted. Mann–Whitney test was used to determine significance. Data were analyzed using Prism GraphPad software.

# Data availability

RNAseq data are available at Gene Expression Omnibus (GSE129366).

**Expanded View** for this article is available online.

## Acknowledgements

We are grateful to Patrick Brennan for advice and technical assistance. S.K.D. was funded by the Bill and Melinda Gates Foundation and the Melanoma Research Alliance, and is a Pew-Stewart Scholar in Biomedical Research. E.C-T was funded by NIH T32CA207021. G.Z.C was funded by the Harvard College PRISE program. H-J.J was funded by the DFCI Center for Cancer Immunotherapy Research. M.B.B and N.M.L. were funded by NIH R01 AI113046. N.M.L. was funded by NIH 1F31 AI138353-01.

## Author contributions

EC-T and GZC designed and conducted the experiments, analyzed the data, and helped write the manuscript. H-JJ and LRA performed the RNAseq analysis. KB and SJC conducted the experiments. NML, LL, and MBB contributed to analysis of adipose-resident iNKT cells. SKD designed the experiments, analyzed the data, and wrote the article with input from all of the authors.

## Conflict of interest

The authors declare that they have no conflict of interest.

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
