## [Review Process File · The EMBO Journal]

Transnuclear mice reveal Peyer's patch iNKT cells that regulate B cell class switching to IgG1

Eleanor Clancy-Thompson, Gui Zhen Chen, Nelson M. LaMarche, Lestat R. Ali, Hee-Jin Jeong, Stephanie J. Crowley, Kelly Boelaars, Michael B. Brenner, Lydia Lynch, and Stephanie K. Dougan

Review timeline:	Submission date:	29th Nov 2018
	Editorial Decision:	4th Jan 2019
	Revision received:	4th Apr 2019
	Editorial Decision:	24th Apr 2019
	Revision received:	28th Apr 2019
	Accepted:	2nd May 2019

Editor: Karin Dumstrei

Transaction Report:

1st Editorial Decision

4th Jan 2019

Thank you for submitting your manuscript to The EMBO Journal. Your study has now been seen by three referees and their comments are provided below.

The referees find the characterization of a novel a population of iNKT cells in the Payer's patches (PP) interesting, but also find that some further analysis is needed for publication here. In particular they would like to see some further insight/support for that PP iNKT cells support PP IgG1+ B cells and their IgG1 secretion via IL-4 production and how PP iNKT cells develop. Should you be able to extend the analysis along the lines suggested by the referees then I would like to invite you to submit a revised version of the manuscript.

I should add that it is EMBO Journal policy to allow only a single major round of revision and that it is therefore important to address the raised points at this stage.

When preparing your letter of response to the referees' comments, please bear in mind that this will form part of the Review Process File, and will therefore be available online to the community. For more details on our Transparent Editorial Process, please visit our website:
http://emboj.embopress.org/about#Transparent_Process

REFeree REPORTS:

Referee #1:

In the current manuscript Clancy-Thompson and colleagues take advantage of transnuclear mice to analyse the phenotype of iNKT cells resident in different tissues. The authors discover a previously uncharacterised population of iNKT cells present in the murine Peyer's patches, which show a unique transcriptional program and produce high levels of IL-4 which seems to be important for the

secretion of IgG1 by intestinal B cells.

While some of the observations presented in the manuscript are interesting, I feel that the paper is somehow disjointed. The first 2 figures are focussed in the characterisation of iNKT transnuclear mice which has been previously published by the authors (Clancy-Thompson et al; J Immunol 2017). Then the manuscript turns towards the characterization of PP iNKT cells. While the characterization of this population is interesting, very few mechanistic insights regarding their generation and/or their function are provided in this manuscript and their link with IgG1 class-switch remains correlative.

Major comments/questions:

(1) The authors suggest that IL-4 production by iNKT cells is responsible for B cell IgG1 class-switch but I don't think that this has been formally demonstrated in the manuscript. Moreover, some of the results are difficult to interpret as increased numbers of iNKT cells or IL-4 production don't seem to correlate with increased IgG1.

For instance, after immunization (Fig 5) there are no differences in IgG1 production between B6 and Va14 mice despite the fact that Va14 mice have 10-40 times more iNKT cells in their PP and mLN (and consequently much more IL-4; Fig 1). On the other hand, in B6 mice there are high numbers of IL-4 producing T cells in the mLN and PP, so one would expect that in Ja18-ko mice there would be some IgG1+ B cells and IgG1 in feces but there seem to be virtually none (Fig 3).

These results are difficult to interpret and the link between IL-4 production by iNKT cells and IgG1 class-switch remains an observation and would require further mechanistic investigation.

(2) The authors say that "PP-NKT provide indirect help to B cells". Whether there any direct interactions between NKT and B cells or the effect seen in IgG1 is only due to indirect NKT cell help has not been experimentally tested

(3) The results of Fig 3 are interesting, but I find very surprising that Ja18KO mice have no IgG1 in feces or IgG1+ B cells in PP or mLN. From the methods is not clear which Ja18-KO mice the authors have used in these experiments. Since the "old" most-widely used Ja18-KO strain is known to have lower TCR repertoire diversity (Bedel et al; Nat Immunol 2012), I wonder whether their phenotype is due to this. The authors should measure IgG1+ B cells in CD1d-KO mice to confirm this phenotype.

(4) In Fig 6D, no stats are provided; but the authors claim that PP iNKTs robustly produce IL-4 after oral aGal. According to the Figure only ~10-15% of PP-iNKT produce IL-4 after oral aGal vs 5-8% of cells in iv administration. This seems quite a small proportion of cells being activated particularly if, as the authors suggest, iNKT-derived IL-4 is responsible for all of the IgG1 production in the gut in steady-state conditions (Fig 3).

(5) In Fig 1 the authors assume that the cytokines detected in their culture are secreted by iNKT cells. Since the experiments are performed with "mixed" cultures of cells from the various tissues it is impossible to determine which cells are producing which cytokines. As there are more cytokines being produced in Va14 cultures than in Ja18 cultures the authors conclude that cytokines are produced by iNKT cells. This is not necessarily the case as it is well established that cytokines secreted by iNKT cells control cytokine secretion by other immune cells.

(6) Related to (5), there seem to be some differences in the cytokine secretion of the various transnuclear mice (supplementary Figure 1; e.g. splenic Vb7c cultures have little IL-10 in comparison with Vb7a cultures). Whether these differences are significant or the mechanisms underlying them (e.g. iNKT cells numbers vs TCR affinity) are unclear and not discussed.

(7) In Fig 7 the authors show a tissue-residence signature present in PP iNKT cells but not in splenic cells. This is surprising as is well established that splenic iNKT cells are tissue-resident cells.

Minor comments:

Statistical analyses are missing in some figures and should be revised

Referee #2:

Clancy-Thompson et al investigated the effector profiles of tissue resident CD1d-restricted iNKT cells in a unique model generated by transnuclear (TN) expression of a set of rearranged TCRs from syngeneic cells. Over-represented TN iNKT cells derived from several tissues display, by and large, the same effector phenotypes of wt counterparts, suggesting that the transgenic expression of the monoclonal TCRs do not skew either peripheral migration or tissue-specific effector functions, typical of mouse iNKT cells. Interestingly, the iNKT cells expansion in TN mice reveals an elusive population of cells resident in Peyer's patches (PP), which can be then detected also in wt animals, and express a distinct effector phenotype from any other tissue resident iNKT cells, including strong IL4 production (>than IFN γ). The presence of these iNKT cell subset associates with an expansion of PP-IgG1+ B cells and the secretion of fecal IgG1, which can be boosted by oral administration of a protein vaccine formulated with the potent iNKT cells agonist aGalCer. Hence, this study discloses a new potential and interesting subset of intestinal iNKT cells that may have a role in mucosal protection.

Major points

1. Unlike the authors claim, there study does not provide any experimental evidence that PP iNKT cells "indirectly" help/support PP IgG1+ B cells and their IgG1 secretion via IL-4 production. It is only show that PP iNKT cells produce IL-4, and this temporally correlates with PP IgG1. Hence, it would be important to substantiate the claim by showing that the iNKT cells help to PP-IgG1+ B cells is: i. CD1d-independent (non-cognate); ii. IL-4 dependent. Can the authors exploit TN PP-derived iNKT cell transfer into CD1dko mice, or the transfer of the same cells into Ja18ko mice plus blocking anti-IL mAbs and, in either transfer models, do oral vaccination with Ova+aGalCer?
2. How do the PP iNKT cells develop: are they already present in the thymus of TN mice, or they are rather a peripheral product. Can TN iNKT cells from the spleen (or other organs) home into PP of Ja18ko mice and give raise locally to the "autochthonous" subset described in TN mice?
3. Concerning Figure 1 results, it would be important to normalize the cytokine secretion in vitro for the actual number of iNKT cells that were plated for the co- culture with the CD1d-expressing APCs.

Minor point

1. Again for Figure 1, the color code utilized to mark the various tissues of iNKT cell origin are somewhat difficult to read.

Referee #3:

This work by Clancy-Thompson et al describes a population of Peyer's patches iNKT cells that are presumably tissue resident, produce IL-4 and impact Ab production by B cells. These cells were revealed in previously generated TN mice, but are also present in B6 mice. The work is clear and well presented. However, the following comments and suggestions should be addressed/considered:

Some references that appear in the core manuscript appear to be missing from the list of reference section. This includes de Ley et al 2018 and Selvanantham et al 2016 references. Please carefully check to avoid omissions.

On page 3, the statement that iNKT cells "help shape the nascent microbiome" is not supported by the An and Olszak references provided. The work from Sáez de Guinoa et al 2018, as well as Selvanantham et al 2016 should be cited. A similar statement also appears in the discussion. In fact, whether iNKT cells can shape the intestinal microbiota has not yet been conclusively and reproducibly shown, and the methods used in these papers (littermate vs. non-littermate, co-housing, single caging, etc.) should be carefully considered.

Add the work from Monteiro et al 2010 re: regulatory iNKT cells.

On page 4, the statement that "Liver iNKT cells adopted more of an NKT1-like profile and produced CXCL9, CXCL10, and IFN-" should be contrasted with the high amounts of IL-4 detected. This

seems to be biologically relevant as recently shown by Liew et al 2017 (should be added). This may also suggest that there is not a strict parallel between the current definition of iNKT cell subsets using transcription factors, and their functional capabilities. This would also be supported by some of the data presented here.

Figure 2 should depict PLZF vs. T-bet and PLZF vs. RORgt dot plots in order to clearly see the 3 iNKT cell subsets, as originally defined by the Hogquist group, and subsequently others. Supplementary figure 2 should show histogram plots of E4BP4. This figure should also show isotype controls or FMOs.

Page 5 line 27, I believe the work refers to Figure 3A, B.

From Figure 5 and Supplementary Figure 4, it is not clear to this reviewer what "after one dose" (page 6 line 33) refers to. Does this refer to d8? If so, supplementary figure 4 does not appear to show Ab production at d8. Please clarify.

Figure 6 should also show PLZF vs. T-bet and PLZF vs. RORgt dot plots. The timing of analyses following intravenous injection or oral gavage appears to be missing from the figure legend and the method section. Please clarify the methods for this experiment. The IL-4 staining is not very convincing. Isotype controls or FMOs should be added to clearly assess IL-4 production by PP iNKT cells.

Figure 7A should include the variance and p value.

PP iNKT cells appear to have a tissue resident transcriptomic profile, which differs from spleen iNKT cells. This may simply due to the sub-anatomical location of the two populations, blood capillaries vs. parenchyma. If so, maybe PP iNKTs should be compared to lung of skin-resident iNKT cells. This should at least be discussed.

The section discussing the Clancy-Thompson 2017 paper on TN mice and "antigen preference" or "ligand specificity" should include the work of Hogquist, Gapin and others on TCR affinity/signal strength (Matulis et al 2010, Lee et al 2013, Cruz Tleugabulova et al 2016, Tuttle et al 2018, Zhao et al 2018). Although unknown differences between experimental approaches (TN mice, retrogenic mice or others) may explain some of the discrepancy, the consensus from this body appear to be that the TCR influences iNKT cell effector differentiation. That being said, this is likely not the only layer of regulation, and it appears that tissue residency also modulates iNKT cell function, which would be really interesting. I believe that a clear discussion around this subject would strengthen the manuscript.

Since a lot of the work rely on TN mice, the authors should briefly describe these mice and how they were generated, especially the Va14 mice that are used throughout the manuscript.

1st Revision - authors' response

4th Apr 2019

Please see next page.

Referee #1:

In the current manuscript Clancy-Thompson and colleagues take advantage of transnuclear mice to analyse the phenotype of iNKT cells resident in different tissues. The authors discover a previously uncharacterised population of iNKT cells present in the murine Peyer's patches, which show a unique transcriptional program and produce high levels of IL-4 which seems to be important for the secretion of IgG1 by intestinal B cells.

While some of the observations presented in the manuscript are interesting, I feel that the paper is somehow disjointed. The first 2 figures are focussed in the characterisation of iNKT transnuclear mice which has been previously published by the authors (Clancy-Thompson et al; J Immunol 2017). Then the manuscript turns towards the characterization of PP iNKT cells. While the characterization of this population is interesting, very few mechanistic insights regarding their generation and/or their function are provided in this manuscript and their link with IgG1 class-switch remains correlative.

Although we previously published the iNKT transnuclear mice, we did not report iNKT cells from adipose or lung, which we feel are an important addition as these (particularly adipose) iNKT cells represent functionally distinct populations not encompassed by the typical NKT1/2/17 categories. The side-by-side comparison to B6 mice is important for establishing the fidelity of the transnuclear model for studying rare tissue-resident iNKT cell populations, which then led us to look in Peyer's patches. We have revised the text to make this point more clearly and flow better with the subsequent figures.

Major comments/questions:

(1) The authors suggest that IL-4 production by iNKT cells is responsible for B cell IgG1 class-switch but I don't think that this has been formally demonstrated in the manuscript. Moreover, some of the results are difficult to interpret as increased numbers of iNKT cells or IL-4 production don't seem to correlate with increased IgG1.

For instance, after immunization (Fig 5) there are no differences in IgG1 production between B6 and Va14 mice despite the fact that Va14 mice have 10-40 times more iNKT cells in their PP and mLN (and consequently much more IL-4; Fig 1).

The oral immunization with α GalCer results in Figure 5 are consistent with the results from Figure 3 showing that B6 and iNKT TN mice have similar fecal IgG1 titers and similar frequencies of IgG1+ B cells in mLN and Peyer's patches. We interpret these data to indicate that a minimum threshold of NKT cells is adequate for IgG1 class switching. This is true for both steady-state and post-vaccination IgG1.

In Supplemental Figure 3 we showed that iNKT TN mice had dramatically higher levels of IgG1+ B cells in PP as compared to littermate control mice. This experiment was performed in an animal facility that no longer exists, so we cannot evaluate the specific IgG1 response to oral immunization in this setting.

On the other hand, in B6 mice there are high numbers of IL-4 producing T cells in the mLN and PP, so one would expect that in Ja18-ko mice there would be some IgG1+ B cells and IgG1 in feces but there seem to be virtually none (Fig 3).

Figure 4 shows ex vivo cultured cells stimulated with PMA/ION to evaluate which cells are poised to make IL-4. Few overall cells were IL-4 positive (range= 0.35-4.66% of total CD3+). The majority of the IL-4 was derived from CD1d tetramer+ iNKT cells. However, this experiment does not demonstrate that iNKT cells are the major source of IL-4 in vivo. To demonstrate that IL-4 in vivo is being produced by iNKT cells, we used in vivo administration of Brefeldin A as shown in Figure 6.

IgG1 is not a major component of fecal Ig, and its presence may be more reliant on iNKT cells. Both serum levels of IgG1 and IgG1+ B cells in the spleen were at normal levels in $J\alpha 18^{-/-}$ mice, indicating that the defect in IgG1 class switching is localized to the gut.

These results are difficult to interpret and the link between IL-4 production by iNKT cells and IgG1 class-switch remains an observation and would require further mechanistic investigation.

(2) The authors say that "PP-NKT provide indirect help to B cells". Whether there any direct interactions between NKT and B cells or the effect seen in IgG1 is only due to indirect NKT cell help has not been experimentally tested

We have now addressed this question using in vitro co-cultures of PP iNKT cells with WT or CD1d $^{-/-}$ B cells with or without blocking antibodies to IL-4. IgM production was not dependent on iNKT cells; however, IgG1 production required iNKT cells and was blocked by addition of anti-IL4. CD1d expression by B cells was not required to receive iNKT cell help. These new data are presented as Revised Figure 4D-G and Supplemental Figure 4.

(3) The results of Fig 3 are interesting, but I find very surprising that Ja18KO mice have no IgG1 in feces or IgG1+ B cells in PP or mLN. From the methods is not clear which Ja18-KO mice the authors have used in these experiments. Since the "old" most-widely used Ja18-KO strain is known to have lower TCR repertoire diversity (Bedel et al; Nat Immunol 2012), I wonder whether their phenotype is due to this. The authors should measure IgG1+ B cells in CD1d-KO mice to confirm this phenotype.

The reviewer is correct that our $J\alpha 18^{-/-}$ mice have potentially restricted TCR repertoire diversity. We have now cited this reference, and also measured IgG1 titers in feces of CD1d $^{-/-}$ mice. Similar to $J\alpha 18^{-/-}$ mice, CD1d $^{-/-}$ show reduced fecal IgG1, and reduced IgG1+ B cells in mLN and PP. These new data are presented in Revised Supplemental Figure 3.

(4) In Fig 6D, no stats are provided; but the authors claim that PP iNKTs robustly produce IL-4 after oral aGal. According to the Figure only ~10-15% of PP-iNKT produce IL-4 after oral aGal vs 5-8% of cells in iv administration. This seems quite a small proportion of cells being activated particularly if, as the authors suggest, iNKT-derived IL-4 is responsible for all of the IgG1 production in the gut in steady-state conditions (Fig 3).

We have added statistics to Figure 6D. We think only a small amount of IL-4 is needed for IgG1 class switching, which is also why the few iNKT cells present in Peyer's patches of B6 mice are sufficient to induce IgG1 class switching to a similar extent as B6 mice post oral vaccination with α -GalCer. We have rephrased the description of these results to remove the word "robustly" and to more accurately describe the minimum threshold number of iNKT cells hypothesis.

(5) In Fig 1 the authors assume that the cytokines detected in their culture are secreted by iNKT

cells. Since the experiments are performed with "mixed" cultures of cells from the various tissues it is impossible to determine which cells are producing which cytokines. As there are more cytokines being produced in Va14 cultures than in Ja18 cultures the authors conclude that cytokines are produced by iNKT cells. This is not necessarily the case as it is well established that cytokines secreted by iNKT cells control cytokine secretion by other immune cells.

We designed these experiments to address the coordinating function of iNKT cells, which includes activating other cell types to produce cytokines. The reviewer is correct that the cytokines are dependent on - but not necessarily coming from - iNKT cells. We have added the following explanation to the results section for clarity:

"Unfractionated lymphocyte populations were used, thus the cytokines analyzed were not necessarily secreted by iNKT cells directly. To determine which cytokines and chemokines were produced in an iNKT cell-dependent manner, lymphocytes from $J\alpha 18^{-/-}$ mice stimulated with α -GalCer were included as a negative control. As a second negative control, $V\alpha 14$ lymphocytes were cultured in the absence of added antigen to determine the production of iNKT-dependent cytokines in response to endogenous ligands."

(6) Related to (5), there seem to be some differences in the cytokine secretion of the various transnuclear mice (supplementary Figure 1; e.g. splenic Vb7c cultures have little IL-10 in comparison with Vb7a cultures). Whether these differences are significant or the mechanisms underlying them (e.g. iNKT cells numbers vs TCR affinity) are unclear and not discussed.

The contribution of the iNKT TCR to iNKT cell functional subsets was the subject of our previous publication (Clancy-Thompson et al. Journal of Immunology 2017). We showed that the contribution of the TCR, albeit significant, was fairly minimal compared to the influence of the tissue of origin. In Supplementary Figure 1, cultures were not normalized to iNKT cell number, but overall we think that if differences related to the TCR are present, they are subtle compared to the tissue of origin. We have added a qualifying statement as to this effect in the results section. All other experiments were conducted using polyclonal iNKT cells and $V\alpha 14$ iNKT TN mice.

(7) In Fig 7 the authors show a tissue-residence signature present in PP iNKT cells but not in splenic cells. This is surprising as is well established that splenic iNKT cells are tissue-resident cells.

Spleen iNKT cells show low rates of chimerism in parabiosis studies, which defines them as tissue-resident. However, other parameters of tissue residency, such as CD103/CD69 coexpression do not necessarily apply to spleen iNKT cells. In a recent paper by Olivier Lantz's group, spleen iNKT cells showed parabiosis chimerism of 5-10%, yet only 6% of these cells expressed CD103. Furthermore the same study showed that blockade of LFA1/ICAM1 integrins resulted in loss of splenic iNKT cells. In our study, we compared spleen iNKT to PP-NKT and showed that the PP-NKT cells had a higher expression of tissue-residency genes CD103 and CD69, but that the splenic iNKT cells were still 20-30% positive for CD103, consistent with previous literature. We now cite the Lantz 2019 paper.

Minor comments:

Statistical analyses are missing in some figures and should be revised

We have added statistics where they were missing in Figures 2 and 6.

Referee #2:

Clancy-Thompson et al investigated the effector profiles of tissue resident CD1d-restricted iNKT cells in a unique model generated by transnuclear (TN) expression of a set of rearranged TCRs from syngeneic cells. Over-represented TN iNKT cells derived from several tissues display, by and large, the same effector phenotypes of wt counterparts, suggesting that the transgenic expression of the monoclonal TCRs do not skew either peripheral migration or tissue-specific effector functions, typical of mouse iNKT cells. Interestingly, the iNKT cells expansion in TN mice reveals an elusive population of cells resident in Peyer's patches (PP), which can be then detected also in wt animals, and express a distinct effector phenotype from any other tissue resident iNKT cells, including strong IL4 production (>than IFN γ). The presence of these iNKT cell subset associates with an expansion of PP-IgG1+ B cells and the secretion of fecal IgG1, which can be boosted by oral administration of a protein vaccine formulated with the potent iNKT cells agonist α GalCer. Hence, this study discloses a new potential and interesting subset of intestinal iNKT cells that may have a role in mucosal protection.

Major points

1. Unlike the authors claim, their study does not provide any experimental evidence that PP iNKT cells "indirectly" help/support PP IgG1+ B cells and their IgG1 secretion via IL-4 production. It only shows that PP iNKT cells produce IL-4, and this temporally correlates with PP IgG1. Hence, it would be important to substantiate the claim by showing that the iNKT cells help to PP-IgG1+ B cells is: i. CD1d-independent (non-cognate); ii. IL-4 dependent. Can the authors exploit TN PP-derived iNKT cell transfer into CD1dko mice, or the transfer of the same cells into Ja18ko mice plus blocking anti-IL mAbs and, in either transfer models, do oral vaccination with Ova+ α GalCer?

As described below, adoptively transferred iNKT cells do not repopulate the Peyer's patches, which makes this question difficult to ask *in vivo*. We opted to use a reductionistic *in vitro* model of B cell class switching instead. We cocultured wild type or CD1d $^{-/-}$ B cells with V α 14 iNKT cells and blocking antibodies to IL-4. iNKT cells were obtained from inguinal LN, mesenteric LN, or Peyer's patches. IL-4 $^{-/-}$ Peyer's patch cells were used as a negative control. We show that the frequency of IgG1+ class-switched B cells is dependent on IL-4 while IgA is not. Furthermore, we show that CD1d expression on B cells is not required to receive iNKT cell help for IgG1 class switching. These data are now presented in Revised Figure 4D-G and Supplemental Figure 4.

2. How do the PP iNKT cells develop: are they already present in the thymus of TN mice, or they are rather a peripheral product. Can TN iNKT cells from the spleen (or other organs) home into PP of Ja18ko mice and give rise locally to the "autochthonous" subset described in TN mice?

This is a difficult question to ask, as PP-iNKT are not defined by a canonical transcription factor that would facilitate their identification in the thymus. We performed several adoptive transfers to determine whether iNKT cells could seed the PP in adult mice. Although transferred iNKT cells could be detected in spleen, liver, adipose tissue and lymph nodes, we did not detect transferred iNKT cells in Peyer's patches of irradiated Ja18 $^{-/-}$ mice post transfer. Even in the tissues where iNKT cells were present, they appear to have altered functional capacity, with no production of IL-4 above background levels after restimulation *ex vivo* with α -GalCer. We have included these data as Supplemental Figure 7.

3. Concerning Figure 1 results, it would be important to normalize the cytokine secretion *in vitro* for the actual number of iNKT cells that were plated for the co-culture with the CD1d-expressing

APCs.

Given the important role of iNKT cells in coordinating cytokine secretion by other cell types, we felt that including the entire leukocyte fraction for each organ was more informative than isolating purified iNKT cells. Normalizing to iNKT cell frequencies would have resulted in different total numbers of cells per condition, which would also have skewed the data. We have better explained our experimental setup and rationale by including the following statements in the results text:

“Unfractionated lymphocyte populations were used, thus the cytokines analyzed were not necessarily secreted by iNKT cells directly. To determine which cytokines and chemokines were produced in an iNKT cell-dependent manner, lymphocytes from $J\alpha 18^{-/-}$ mice stimulated with α -GalCer were included as a negative control. As a second negative control, $V\alpha 14$ lymphocytes were cultured in the absence of added antigen to determine the production of iNKT-dependent cytokines in response to endogenous ligands.”

Minor point

1. Again for Figure 1, the color code utilized to mark the various tissues of iNKT cell origin are somewhat difficult to read.

We changed the thymus from yellow to black and have reoriented the legend to be more clearly legible.

Referee #3:

This work by Clancy-Thompson et al describes a population of Peyer's patches iNKT cells that are presumably tissue resident, produce IL-4 and impact Ab production by B cells. These cells were revealed in previously generated TN mice, but are also present in B6 mice. The work is clear and well presented. However, the following comments and suggestions should be addressed/considered:

Some references that appear in the core manuscript appear to be missing from the list of reference section. This includes de Ley et al 2018 and Selvanantham et al 2016 references. Please carefully check to avoid omissions.

Selvanantham et al 2016 is present. We could not identify a de Ley 2018 reference in PubMed, although we are happy to include it if the reviewer could provide the reference via the journal editor.

On page 3, the statement that iNKT cells "help shape the nascent microbiome" is not supported by the An and Olszak references provided. The work from Sáez de Guinoa et al 2018, as well as Selvanantham et al 2016 should be cited. A similar statement also appears in the discussion. In fact, whether iNKT cells can shape the intestinal microbiota has not yet been conclusively and reproducibly shown, and the methods used in these papers (littermate vs. non-littermate, co-housing, single caging, etc.) should be carefully considered.

We have added the new reference, correctly cited the An and Olszak references to not refer to NKT cells acting on the microbiome, and rephrased the intro and discussion sections.

Add the work from Monteiro et al 2010 re: regulatory iNKT cells.

We have added this reference.

On page 4, the statement that "Liver iNKT cells adopted more of an NKT1-like profile and produced CXCL9, CXCL10, and IFN-" should be contrasted with the high amounts of IL-4 detected. This seems to be biologically relevant as recently shown by Liew et al 2017 (should be added). This may also suggest that there is not a strict parallel between the current definition of iNKT cell subsets using transcription factors, and their functional capabilities. This would also be supported by some of the data presented here.

Thank you, we have added this point and reference.

Figure 2 should depict PLZF vs. T-bet and PLZF vs. ROR γ t dot plots in order to clearly see the 3 iNKT cell subsets, as originally defined by the Hogquist group, and subsequently others. Supplementary figure 2 should show histogram plots of E4BP4. This figure should also show isotype controls or FMOs.

We have added E4BP4 histograms, complete with isotype control staining to Supplemental Figure 2. We revised Figure 2 to show PLZF on the y-axis and either T-bet or ROR γ t on the x axis. For clarity, we have revised the bar graphs to quantify NKT1/2/17 cells as shown in these reformatted gates, added statistics, and revised the text describing these results.

Page 5 line 27, I believe the work refers to Figure 3A, B.

Thank you. We made this correction.

From Figure 5 and Supplementary Figure 4, it is not clear to this reviewer what "after one dose" (page 6 line 33) refers to. Does this refer to d8? If so, supplementary figure 4 does not appear to show Ab production at d8. Please clarify.

Thank you for noticing this. We meant day 15 and have now clarify the text to read "after one boost (Day 15)".

Figure 6 should also show PLZF vs. T-bet and PLZF vs. ROR γ t dot plots. The timing of analyses following intravenous injection or oral gavage appears to be missing from the figure legend and the method section. Please clarify the methods for this experiment. The IL-4 staining is not very convincing. Isotype controls or FMOs should be added to clearly assess IL-4 production by PP iNKT cells.

We have added experimental details to the figure legend and methods section. The mice receiving no α -GalCer were used as a negative control to set the gates for IL-4 production, and statistics are now included to compare IL-4 production between oral α -GalCer and no α -GalCer groups.

Figure 7A should include the variance and p value.

The PCA plot in Figure 7A shows % of variance along principal components 1 and 2. The total variance is 1128.203, which we have included in the methods section.

PP iNKT cells appear to have a tissue resident transcriptomic profile, which differs from spleen

iNKT cells. This may simply due to the sub-anatomical location of the two populations, blood capillaries vs. parenchyma. If so, maybe PP iNKTs should be compared to lung of skin-resident iNKT cells. This should at least be discussed.

We now include in the discussion a 2019 reference from Olivier Lantz's group that compared signatures of tissue residency between spleen and lung iNKT cells, which showed similar results to our comparison of spleen and PP-NKT cells.

The section discussing the Clancy-Thompson 2017 paper on TN mice and "antigen preference" or "ligand specificity" should include the work of Hogquist, Gapin and others on TCR affinity/signal strength (Matulis et al 2010, Lee et al 2013, Cruz Tleugabulova et al 2016, Tuttle et al 2018, Zhao et al 2018). Although unknown differences between experimental approaches (TN mice, retrogenic mice or others) may explain some of the discrepancy, the consensus from this body appear to be that the TCR influences iNKT cell effector differentiation. That being said, this is likely not the only layer of regulation, and it appears that tissue residency also modulates iNKT cell function, which would be really interesting. I believe that a clear discussion around this subject would strengthen the manuscript.

We have added these references and additional text to the introduction.

Since a lot of the work rely on TN mice, the authors should briefly describe these mice and how they were generated, especially the Va14 mice that are used throughout the manuscript.

We have added additional description about the V α 14 TN line to the start of the results section. We described the technical aspects of somatic cell nuclear transfer in Clancy-Thompson et al 2017, Dougan 2012, Dougan 2013a and Dougan 2013b.

2nd Editorial Decision

24th Apr 2019

Thanks for submitting your revised manuscript to The EMBO Journal. Your study has now been re-reviewed by referees #1 and 2. As you can see below both referees appreciate the introduced changes and support publication here.

I am therefore very pleased to let you know that we will accept the manuscript for publication here. Before I can send you the formal accept letter - there are just a few editorial things we need to sort out. You can use the link below to upload the revised version.

REFeree REPORTS:

Referee #1:

In this revised version of their manuscript, the authors have addressed most of my comments and concerns originated from their original submission. The in vivo link between IgG1 and NKT-derived IL-4 remains correlative, but the new in vitro experiments point towards a CD1d-independent mechanism controlling IgG1 levels. The manuscript is significantly improved by the new experiments and the changes in the text.

Referee #2:

I am satisfied with the answers authors have provided to my queries. The study has been substantially improved by the newly added experiments and comments.

2nd Revision - authors' response

28th Apr 2019

The authors performed all requested editorial changes.

Corresponding Author Name: Stephanie Dougan

Manuscript Number: EMBOJ-2018-101260R